# RGB Color Model: Effect of Color Change on a User in a VR Art Gallery Using Polygraph

**DOI:** 10.3390/s24154926

**Published:** 2024-07-30

**Authors:** Irena Drofova, Paul Richard, Martin Fajkus, Pavel Valasek, Stanislav Sehnalek, Milan Adamek

**Affiliations:** 1Faculty of Applied Informatics, Tomas Bata University in Zlín, 760 05 Zlín, Czech Republic; 2Polytech Angers, University of Angers, 49 000 Angers, France; paul.richard@univ-angers.fr

**Keywords:** color model, gamut, art gallery, virtual reality, art digitization, digital twins, artwork, 3D model, lie detector, polygraph, sensors, signals

## Abstract

This paper presents computer and color vision research focusing on human color perception in VR environments. A VR art gallery with digital twins of original artworks is created for this experiment. In this research, the field of colorimetry and the application of the L*a*b* and RGB color models are applied. The inter-relationships of the two color models are applied to create a color modification of the VR art gallery environment using C# Script procedures. This color-edited VR environment works with a smooth change in color tone in a given time interval. At the same time, a sudden change in the color of the RGB environment is defined in this interval. This experiment aims to record a user’s reaction embedded in a VR environment and the effect of color changes on human perception in a VR environment. This research uses lie detector sensors that record the physiological changes of the user embedded in VR. Five sensors are used to record the signal. An experiment on the influence of the user’s color perception in a VR environment using lie detector sensors has never been conducted. This research defines the basic methodology for analyzing and evaluating the recorded signals from the lie detector. The presented text thus provides a basis for further research in the field of colors and human color vision in a VR environment and lays an objective basis for use in many scientific and commercial areas.

## 1. Introduction

This document presents research on color vision in a virtual reality (VR) environment. Currently, color vision, as well as computer graphics and computer vision, is an integral part of information technology issues. We encounter computer graphics in almost all fields. At the same time, the type of method of image processing used differs in each field and individual graphics subdisciplines. With 3D graphics and three-dimensional space development, new approaches and solutions have emerged in this field. The basic principles of 2D graphics and analog technologies are smoothly reflected in digital 3D image processing [1]. Awareness of these procedures makes it easier to work with images, colors, and materials so that they fully correspond to the perception of the environment by the human eye [2]. All these principles are used today in the game industry, engineering, design, architecture, art, therapy, military, criminology, agriculture, psychology, chemistry, and other fields [3,4,5].

Currently, environments are usually created and simulated in 3D space or virtual or augmented reality [6]. For the human eye to perceive virtual objects, scenes, and environments realistically, the science of colors, color models, color spaces, light radiation, the anatomy of the human eye, and Human vision is fully applied [7]. These issues are summarized and described in more detail by the Colorimetry and Color management field. This is a vast field of science. Colorimetry includes the color calibration of display devices, issues of printing, light, sensing devices, analog and digital processing of image data and signals, color mixing, digital description of colors, tone, color psychology, and more [7,8]. The issue of Colorimetry and Color vision is currently the subject of extensive research in many scientific fields.

This manuscript describes the basic principles of mixing primary and secondary colors. These basic principles are experimentally applied in a virtual reality (VR) environment to observe the influence of colors and color changes on human color perception. The display and perception of colors in a VR environment have already been the subject of some research [9]. However, the experiment described in this paper has not yet been performed. For these purposes, an art gallery was created in a virtual environment. In this virtual reality environment, real works of art by the painter Michal Pasma, which have been digitized, are presented. The experiment aims to capture the response to color changes of a user embedded in a virtual gallery environment. A lie detector was used to identify changes in color perception in the environment. Several scientific research experiments have already been conducted with this device. They are mainly in the field of human psychology and perception [10,11,12].

This experiment explores the possibilities of working with color vision and the RGB (Red, Green, Blue) color model in a VR environment. This experiment could be the basis for further research in the field of Computer vision, Color vision, and the effect of the environment on the user embedded in VR. Practical use is possible in medicine, psychology, therapies, or educational programs, as well as in commercial fields. It is not known whether a similar experiment has already been implemented and the issue of color vision applied in a VR environment using a lie detector for data analysis.

This primary research is interdisciplinary. It connects the scientific fields of informatics, art, and psychology [13,14]. Due to the vast possibilities of using VR, research on the effect of the VR environment on its users is essential. The created VR environment can significantly impact its users regarding cognitive perception. Like the colors that apply in an environment, design and purpose can fundamentally influence the user’s emotions. Ultimately, this simulation process can have an impact not only on the user’s perception of art but also a significant impact on the art trade. The choice of color layout of the virtual environment has a considerable influence on the presentation of art collections and the individual experience of art presentation. Colors significantly affect human psychology and the well-being of the user. Nowadays, color therapy is also becoming more and more popular. In this regard, the environment created in this way can be an excellent tool for observing and developing these therapies and positively impacting their users.

Sensors of a polygraph were used in this experiment. These sensors measure the user’s physiological changes. Polygraphs are mainly used to determine the effect of stressful situations [15,16]. Their application can be found in psychology, the military, aviation, and other industries where a high level of psychological and physical stress is expected. Polygraphs are suitable for simulations and training in stressful situations. In this way, the effect of the color environment on the user embedded in VR can also be detected and measured. However, there is no assumption of an extremely high degree of deviation in measuring individual sensors. Therefore, in this basic research, the primary colors of the RGB model, which were added to the variable environment of the VR art gallery, were measured. It can also be assumed that subtle changes in color tones during the immersion and experience of a VR art presentation can also be measured this way. The influence of colors on the human psyche can thus be investigated in a VR environment, together with other procedures and methods. There is also a prerequisite for application in optometry about individual human color perception. A valuable tool for research and development in this field can be obtained in the scientific field of Colorimetry, together with the study and simulation of lighting conditions.

## 2. Methods of Creating a VR Art Gallery

Methods in the issue of digitizing objects, especially works of art, are derived from work with colors. The digitization of art and reproduction of works of art in a digital, online, or VR environment emphasizes the realistic reproduction of color, material, and structure [17,18,19]. These attributes were also considered when creating realistic digital twins of the artworks used for this experiment. Emphasis was therefore placed primarily on the accurate color production of paintings painted using the technique of acrylic paints on canvas. ColorChecker Classic from X-Rite s.r.o. Vyškov, Czech Republic was used for realistic digital reproduction [19]. The RGB color model was mainly used to modify the color environment of the virtual art gallery. Digital display devices use the RGB color model. The three primary colors of this model (RGB) in absolute values are the source for measuring the effect of colors on users in this study. As part of the color modification of the VR environment, there were transitions of the colors Green Lima (G/Lima), Magenta (M), Magenta Plum (M/Plum), Magenta Purple (M/Purple), Blue Purple (B/Purple), and White (W). In the case of measuring the influence of color tonal changes, the application of the L*a*b* color model is necessary. However, this study is focused on analyzing the influence of the absolute values of the primary colors of the RGB color model.

### 2.1. Colorimetry: Color Model and Color Space

This experiment primarily involves working with the RGB color model on the issue of color models and color spaces (gamuts) [17]. This model is chosen because of digital outputs in the VR environment and the nature of measuring the effect of color changes on the user. The subject of this experiment is measuring the user’s reaction to the absolute color values of all primary colors of the color model (RGB). In connection with these absolute color values, the user’s response to these colors in the form of physiological changes is assumed. In the case of absolute values of R, G, B colors, it is not necessary to consider the color tone. The study must consider the color spaces in which digital and display devices work. The standardized ColorChecker Classic color scale is used for the realistic color reproduction of digitized works of art. The Oculus Quest2 VR display device is used in this experiment. Its default color space is Rec.2020 (HDR) with a white point of D65. In the issue of VR imaging, the color space is not well defined. It may vary depending on the device, the display technology, or the application used. However, the Rec.709 color space is the most common for internet and non-HDR content. In connection with the different range of color space for sensing and display devices, it is also appropriate to work with the L*a*b* color space (CIE 1976), which is derived from the CIE XYZ color space (CIE 1931) [20]. This color space is conceived as independent of the specific device used. The color models and gamuts are shown in Figure 1 and Figure 2.

Figure 1 and Figure 2 show significant differences between color models and gamuts, especially when interpreting the colors of VR display devices. This study focuses primarily on the user’s response in a VR environment to the absolute primary colors RGB. This color model is used in the next steps of creating the VR environment. Measuring the effect of color in this environment is based on color modification. Therefore, additional colors are defined to smoothly transition the colors in the time interval described in Table 1.

It can be assumed that the L*a*b* color model will also be suitable for the analysis and definition of individual color tones during the interval of color modifications based on color tones in the VR environment of the art gallery.

### 2.2. Creating the Digital Twins of the Artworks

A digitized image of actual works of art is used in this study. The art paintings were photographed with a Pentax K-50 DSLR camera. The image was taken in an art gallery room without artificial lighting. The room was lit by natural daylight. The captured digital image was edited in the 2D software Camera RAW 14.5 for photo editing. The digital image was edited with brightness, contrast, and exposure functions and then trimmed to the final display, as shown in Figure 3. Twelve artworks and one information panel about the exhibition’s author were imported into the virtual art gallery. Figure 1 shows the reference artwork (a) in the first capture step and (b) the digitized and post-production edited artwork for application in the VR gallery environment.

The digital image editing procedure is not the subject of this experimental study. The works of art serve primarily as objects intended to focus the user’s attention when embedded in the created VR environment. The RGB color model was used as part of the digital imaging in this image processing.

### 2.3. Creating the Virtual Art Gallery

In the software for creating 3D objects and the VR environment, Unity engine (2021), a room was created in a virtual environment for the experiment’s needs [22]. Realistic art objects by the Czech artist Martin Pasma were presented in this virtual gallery. As Figure 3 shows, the walls of the virtual room have a default color of W (White), which has a value of (255,255,255) in the RGB color model. The color values of the virtual room’s walls change during the user’s embedding into the VR environment, as well as its tone. The purpose of the color modification of the environment is (among other things) to record to what extent and when the user notices a change in the virtual environment. The artistic images presented in the virtual art gallery were crucial in directing and maintaining the user’s immediate attention in the created environment. The user could actively move and study the details of the images up close or from any distance as a complete unit of the presented topic.

The white color of the room’s walls, which highlighted the presented images, was static in the first minute after starting and embedding the user into the VR. After that, the walls began smoothly changing the color value to another tone and color. A range of pre-defined colors was used: Green Lima (G/Lima), Magenta (M), Magenta Plum (M/Plum), Magenta Purple (M/Purple), Blue Purple (B/Purple) a White (W). Figure 4 shows the VR environment and its reference color modifications.

In individual steps indistinguishable to the human eye, these color changes took place over 15 min until the first input value of the wall color W (White), and then the cycle was repeated. However, three sudden intervals with a very contrasting change in the color of the environment were also defined during the cycle as shown in Figure 5a–d. These cycles took place without adjusting the white point and diffuse lighting of the exhibits. The absolute values of the color model R (255), G (255), and B (255) were chosen to measure their influence on the user’s physiological changes in VR. These colors were defined in the environment statically for 10 s. Figure 6 shows the VR environment in R, G, and B background color intervals.

As has been written in this text, the virtual environment of the gallery was created with a color-modifiable background. These color modifications were applied to the walls of the room in the VR environment. The starting background color was W (White), which varied smoothly by one tone across the entire color gamut in a 15 min time cycle. Into this cycle, step-by-step changes to the absolute colors of the RGB color model were applied. These color change modifications aimed to allow observation of the user’s physiological changes when embedded in the VR environment. The lie detector recorded these changes. The following section describes applying color changes in the VR gallery.

### 2.4. Creating C# Script Color Modification

One of the ways to achieve a smooth transition of an object’s color to another color in the Unity engine is by using a C# script. Within the script, we can choose the object, its exact color, and the conditions under which this color will change. This project uses such a script for smooth transitions between 7 different colors. Initially, the desired colors are selected and written in 3 sheets, each containing one of the color components RGB in the same order in which the colors will be displayed and gradually modified:public int[] colR = {0, 255, 147, 221, 255, 106, 255};public int[] colG = {255, 0, 112, 160, 0, 90, 255};public int[] colB = {0, 255, 219, 221, 0, 205, 255};

Also, there is a need to specify 3 other colors:The color that will change (“actualColor” variable).The color it will change to (variables “activeR”, “activeG”, “activeB”).The current color of the object (variables “nextR”, “nextG”, nextB”).

First, using the “activeColor” variable from the sheets with the color values, we call up the value of the color from which we want to start the modification. We save the individual color components in the variables “activeR”, “activeG”, “activeB”:activeR = colR[actualColor];activeG = colG[actualColor];activeB = colB[actualColor];

The color parameters of the next transition are obtained in a similar way. Next, we add 1 to the variable “actualColor”, and the parameters of the following colors are obtained from the lists of color folders. These colors are stored in the variables “nextR”, “nextG” and “nextB”:nextR = colR[actualColor+1];nextG = colG[actualColor+1];nextB = colB[actualColor+1];

In the next steps, the individual “activeX” variables are used to store the transition color parameters. The values of these variables are then numerically approximated to the “nextX” values. A value of 1 is chosen for one step of color approximation.

For each color component, a special method is used, in which it is decided

Whether it is necessary to add or subtract one step to “approach” the next color.Whether the given parameter has already reached the required value.


void calculateR()

{

int split = activeR − nextR;

if (div > 0) {activeR = activeR − 1; doneR = false;}

if (div < 0) {activeR = activeR + 1; doneR = false;}

if (div == 0) {doneR = true;}

tempR = (float)sctiveR/255;

}


Two more variables are also assigned to each color component:
-doneXVariable of type Boolean.Contains information on whether the given color component has already reached the required value. In case the color folder variables “activeX” and “otherX” contain the same values; this variable is set to “true”, otherwise it remains “false”.-tempXVariable of type float.Is used to convert integer variable values to integer variables.

In general, it is customary to use whole numbers in the range 0–255 to express individual color components; however, in the Unity engine, a decimal number with an interval of 0–1 is used for individual color components. Based on the logical continuity of the individual operations, a procedure was chosen where the individual colors between which we want to create a transition are entered in the usual values and the recalculation is only carried out within the running of the script.

Scripts used within the *Unity engine*, by their very nature, allow certain steps to be performed for each newly displayed image on the screen in the form of the “Update ()” method. The commands written in this method are then directly dependent on the number of frames per second (FPSs). As part of the chosen procedure for modifying the color of objects, it is also necessary to determine at what speed the color transition will take place between individual colors and their color components. In this case, two numeric variables “counter” and “interval” were chosen:-CounterVariable, tracking the order of the current frame.-IntervalDetermines how many times the scene is rendered as individual frames before the color components are recalculated and the color of the selected object is updated.

Every time that the “Update()” function is selected, the value of the “counter” variable is increased by 1. If the “contour” variable reaches the value of the “interval” variable, the new values are set in the “tempX” variables; these are applied to the requested object, and the value of the “interval” variable is set to 0 again.


if (counour == interval) {recountR(); recalculateG(); recalculateB();

zed.GetComponent<Renderer>().material.color = new Color(tempR, tempG, tempB); counter = 0;}


Achieving the desired color is detected using the “doneX” variables. Suppose these variables for all color components contain the value “true”; it follows from the logical context that all color components have already reached the values of the color to which the original color was changed, and all variables can be set to values corresponding to the transition to the next color from the list. Otherwise, it is necessary to repeat the individual steps of the selected procedure until the mentioned condition is met.


if (doneR && doneG && doneB) {

actualColor++;

nextR = colR[actualColor + 1];

nextG = colG[actualColor + 1];

nextB = colB[actualColor + 1];

doneR = false;

doneG = false;

doneB = false;

}


## 3. Measurement by Polygraph Sensors and Signals

The embedded user reacts to the VR gallery environment. A model situation was created using lie detector sensors (polygraph). Lie detectors are currently used in many fields, especially criminology, law, and psychology. At the same time, they are also the subject of unmeasured data analysis and evaluation [23,24,25,26,27,28,29]. They usually provide a measurement of changes in physiological properties induced by a specific stimulus. However, an experiment measuring the effect of background color changes on a user embedded in a VR environment has not yet been carried out. Regarding Human color vision, it can be assumed that a sudden change in the color of the virtual environment will be a stimulus for a change in physiological properties. The lie detector LX6 from the Lafayette Instrument POLYGRAPH company, Lafayette at United States was chosen for these purposes [29]. Figure 7 shows a user embedded in a VR environment and connected to sensors to measure changes in physiological properties using a lie detector. In this basic experiment, five test measurements were performed to synchronize and calibrate all devices. The research aimed to determine whether the modification of the colors of the VR environment would affect the user embedded in the VR environment. At the same time, the goal was to determine whether and to what extent polygraph sensors could capture potential physiological changes due to color modification in the environment. Users were tasked with visiting a sale exhibition in a virtual art gallery with the potential to purchase a real work of art from the artist.

Participants did not have any visual impairment, including blindness.Participants had information about the type of VR environment (art gallery with exhibited artworks).Participants were embedded in an active VR environment in which they could move and were introduced to the movement controls using a single joystick.Users had no information about the color modification of the VR environment, but they were introduced to the sensors’ purpose and measurement.Users were immersed in a VR art gallery with classic white walls.There was no noise or disturbing sounds around the user.Verbal responses to the VR environment were recorded in writing.Movements and facial reactions were visually observed during the measurement.

In this basic experiment, no questionnaire or other experimental form was used, except for a subsequent discussion with the participants after the measurement.

Figure 7 shows a user nested in a gallery in a virtual reality environment. The numbers 1 and 3 in Figure 6 define the computing devices used in this experiment. *Rectangle* number 2 shows a device mediating the user’s visual immersion in a virtual art gallery using the Oculus Quest2 VR headset by Meta Platforms, Inc. at United states. The user embedded in the VR environment is simultaneously connected to the sensors of the polygraph. Sensors for measuring physiological changes are marked in Figure 7 in rectangles 4–7 [30,31].

### 3.1. Pneumograph (Sensors 4 and 5)

Two sensors are attached to the user’s body, measuring breathing in the chest and abdominal area. During breathing, air is exchanged during inhalation and exhalation. Regular breathing can be defined by its frequency. This frequency can be determined by the number of breaths per minute. These are respiratory sensors, which are indicated by frames 4 and 5. Sensor 4 detects the frequency of breathing in the area of the upper respiratory tract, and sensor 5 in the vicinity of the diaphragm. These sensors have the task of monitoring changes in lung volume, breathing intensity, and breathing rhythm. The standard breathing rate for healthy adults is 16–20 breaths per minute. Analyzing data from pneumograph sensors is challenging. The user can arbitrarily change the frequency of the breath. Another factor influencing the resulting data can be natural factors, such as coughing, sneezing, swallowing, or talking. A comfortable environment is assumed in this study. A change in breath can be manifested by a reaction to a change in color at a specific time in the entire interval.

### 3.2. Photoelectric Plethysmograph (Sensor 6)

This sensor records rapid changes in pulse blood volume. The sensor records the change through a photosensitive sensor that measures the light reflected or transmitted through the skin. The intensity of the red light falling on the sensor is directly related to the amount of blood. The sensor is, therefore, a blood flow sensor and is shown in Figure 7 in box number 6. This is sometimes referred to as relative blood pressure (RBP), which is what is called “cardiovascular” or the “relative blood volume.” RBPs are shown in the waveforms as two types of signals, blood pressure and pulse change. It is therefore a capture of the amplitude in the duration of the change in blood pressure and the change in pulse waves.

### 3.3. Electrodermal Activity EDA (Sensor 7)

The galvanic skin response (GSR) sensor records a change in the skin’s ability to conduct electricity, where skin conductivity is a prerequisite, and its change is a sign of excitement, nervousness, or psychological pressure. Skin conductance is measured using two stainless steel fingertip electrodes, and sweat gland activity is recorded. In this experiment, the change in skin resistance is measured based on sweat gland activity.

### 3.4. Activity Sensor

This sensor is shown in Figure 7. It is a pressure and movement sensor. The sensor is a chair pad that captures the movement of a seated user embedded in VR. This sensor can also be used on the hands or feet. Figure 7 shows a reference recording of signals from individual sensors, which are listed under the following designations:P1, Abdominal Respiration trace;P2, Thoracic Respiration trace;PL, Photoelectric Plethysmograph;SE, Activity Sensor;GS, Electrodermal Activity (EDA).

Signals from lie detector sensors and their data are processed by software (SW) Lafayette Polygraph System 11.8.5. The processing of signals from all mentioned sensors and the analysis of the measured output data from the lie detector is described in more detail in the following chapter.

## 4. Results

There are many options for evaluating the measured data from the lie detector. The output measured data can be interpreted numerically and also with a graph. The methods used always depend on the purpose and goal of using the lie detector [30,31,32,33,34]. In this experiment, a polygraph (lie detector) was used to record the physiological changes of a user embedded in a VR environment. Changing the colors of the VR environment was supposed to cause these physiological changes in the user. This involved a sudden change in color at a given interval when the user was embedded in a changing environment of a virtual art gallery. Measurements were carried out to detect the physiological changes of the user in the reaction to three absolute process colors RGB in the VR environment. Thus, the values of these R(255)G(255)B(255) absolute colors formed a sudden color change in the background of the virtual gallery.

Recall that the RGB color model does not work with physical print color but with colored light. This is how colors are applied in digital environments, devices, or digital output devices. They were measured using the principles of color and human vision. Responses to these changes were recorded using lie detector sensors, which are described in more detail in the previous chapter. Figure 8 shows the entire time interval of the user’s immersion in the VR gallery environment and the lie detector connected to the sensors. It is not known whether a similar experiment has been conducted in the area of color and human vision in a VR environment. It was, therefore, not clear in advance whether the sensor detectors would detect the user’s physiological changes. There are also no studies to back this experiment up.

Figure 9 shows the total recording of signals from all mentioned sensors in a time interval of 18 min. During this time, the user was embedded in the virtual gallery environment and connected to the sensors of the individual sensors.

Figure 9 shows a recording of the signals of all the mentioned sensors in a time interval of 18 min. The red squares in the lower part of the measured data indicate the time intervals of the step change of the background color to absolute colors. The output of the sensor measurement was a large amount of measured data from all the listed sensors. These data were generated at an interval of 0.03 s. Therefore, the output data had to be analyzed and simplified. For each type of sensor in the total measurement time, the following was determined:Total minimum value ^a^T_min_|Total maximum value ^b^T_max_

These upper and lower bounds of the output recorded data were then defined in the interval of each absolute RGB color, which represented the sudden change in the color environment for 10 s in the VR art gallery in the following order:G(255) G_min_|G_max_; R(255) R_min_|R_max_; B(255) B_min_|B_max_

The following Table 2 lists all the minimum and maximum values obtained from the total amount of measured data. Table 2 also lists the individual sensors according to their designation in the SW Lafayette Polygraph System 11.8.5, as described in Figure 7. Individual sensors are represented by the following labels: P1 Abdominal Respiration trace; P2 Thoracic Respiration trace; PL Photoelectric Plethysmograph; SE Activity Sensor; and GS Electrodermal Activity (EDA).

The following graphic processing in Figure 10 shows these evaluated data for each of the individual colors compared with the minimum and maximum values from the total volume of data.

Figure 10 shows the differences in the output values of signals measured by individual sensors. It can be seen from the graphic representation that the user embedded in the VR environment of the art gallery perceived a sudden change in the color of the environment. The graph shows the values of the sensors P1 Abdominal Respiration trace, P2 Thoracic Respiration trace, PL Photoelectric Plethysmograph, and SE Activity Sensor. The graph in Figure 10 does not show the signal values from the GS Electrodermal Activity (EDA) sensor. According to the values shown in Table 2, it is evident that the values measured by this sensor are significantly lower. For this sensor, a statistical evaluation procedure using a statistical “Time series method” was used, and a *continuous linear trend* was calculated, which is shown in Figure 10 and mathematically described as follows:GS y = 58,461x + 4 × 10^6^(1)

From the waveform of the recorded signal from the GS sensor (EDA) and the linear function shown in Figure 11, it can be seen that a physiological change in the user embedded in the VR environment occurred in the total time interval. Table 3 shows the reaction to the individual absolute values of the RGB colors. The table below shows the differences in the measured values of all sensors in the individual intervals of the color change in the environment in connection with the total interval of the user’s immersion in the VR environment:G(255) G_min_|R(255) R_min_|B(255) B_min_| − (^a^T_min_)
(^b^T_max_) − |G(255) G_max_|R(255) R_max_|B(255) B_max_

The numerical values measured by all mentioned lie detector sensors show the physiological changes of the user in response to a sudden change in color in the virtual gallery environment, as shown in Table 3 and Figure 10 and Figure 11. It is necessary to mention that other attributes affect the entire experiment during the measurement and subsequent analysis of the obtained data. This is a model situation to find out whether the sensors of the lie detector will capture the physiological changes of the user immersed in the VR art gallery. The evaluation of the measured values of the GS sensor is performed differently due to the order-of-magnitude-lower values of the signal recorded by this sensor, as shown in Figure 10. This procedure suggests a possible further evaluation of the entire variable environment in a smooth transition of color tone between intervals of sudden RGB color changes. This issue is described in 3 and 4. For further research, linear functions were determined for all other sensors used in this experiment. Figure 12 and Figure 13 show their graphic representation.
**P1→** y = 2075,7x + 1 × 10^9^(2)
**P2→** y = −44,509x + 2 × 10^9^(3)
**PL→** y = 156,04x + 1 × 10^9^(4)
**SE→** y = −2021,5x + 7 × 10^8^(5)

It can be assumed that the linear trend of the entire signal recording can help define the user’s reactions to individual tones in a color-modified virtual gallery environment, especially in identifying individual points on the recorded curve. This procedure will be applied in future research that builds on this basic experiment. Other attributes that may affect the user’s physiological changes and that are not considered in this practical research are discussed in the next chapter. In future research, attention will also be focused on individual aspects and analysis of work with individual lie detector sensors. However, the practical use of color and human vision in the VR environment described in this experiment can be assumed in many scientific and commercial fields.

## 5. Discussion and Conclusions

This experiment is the first to focus on the field of color vision and human vision in a virtual reality environment from the point of view of the color perception of a user embedded in a VR environment. The text mentions the creation of digital twins of actual works of art. Another topic is the realization of modifying the VR environment in terms of color. In a positive reaction, the stimulation of changes using an experiment to carry out color change in a VR environment was captured by the sensors of a lie detector. There is high potential for different color and computer vision research in digital twin research. This is also related to work with color models and spaces (gamut), which significantly influence the final result of the reproduction of the work of art [35]. HSV and HSB models are closest to human vision and can always be applied [36,37,38]. Nevertheless, using the color model and gamut L*a*b* is more appropriate, as it has the most extensive color range (CIE 1976) and does not consider the color spaces of sensing and display devices. In this regard, a crucial element is the display of the VR headset, which can vary according to the type of headset and other factors mentioned in Section 2. This color model can also be suitable for determining a specific color in connection with the user’s reaction in the case of tonal transitions in the total interval of color modification in the VR environment. Analyzing the reaction to these color tone changes will also be the subject of further research.

The subject of this research is mainly working with the RGB color model, which is used to modify the color environment in VR. This environment contains digital twins of the artworks. In this VR environment, smooth transitions of color tones and sudden changes in the environment to absolute RGB values were created. In the Unity engine, a smooth transition of the object’s color to another color was achieved using a C# Script. The required colors were entered in the form of three sheets. Each of the sheets contained one of the RGB color components in the same order in which these colors were displayed and modified in turn:public int[] colR = {0, 255, 147, 221, 255, 106, 255};public int[] colG = {255, 0, 112, 160, 0, 90, 255};public int[] colB = {0, 255, 219, 221, 0, 205, 255};

To the variable “**currentColor**”, 1 was added, and the parameters of the following colors were obtained from the lists of colors that we stored in the variables “**nextR**”, “**nextG**” and “**nextB**”.

Additional procedures are provided in this chapter. It is necessary to mention that this is one of the options for achieving the need for color modifications. In addition to using the C# Script, you can consider applying animation for color modification in the Unity engine.

A polygraph was chosen to measure a user’s reaction when embedded in a VR environment. This was the premise of recording physiological changes in connection with the reaction to a color change in the environment. As part of a search of professional publications, similar use of a lie detector has not been found to date, as well as the measurement of a human reaction to a color change in a VR environment. Therefore, it was impossible to base the research on previous data in this experiment. So far, the research is mainly aimed at criminology, law, and psychology [39,40]. Although the field of color psychology is the focus of much research, none has yet been conducted in the context of this presented research [40,41,42,43,44]. This research aimed to determine whether the measurement of human perceptions by lie detector sensors in a VR environment can be evaluated and how the measured data can be evaluated.

In this basic experiment, five test measurements were performed to synchronize and calibrate all devices. The research aimed to determine whether the modification of the colors of the VR environment would affect the user embedded in the VR environment. At the same time, the goal was to determine whether and to what extent polygraph sensors could capture potential physiological changes due to color modification in the environment. Ten volunteers aged 25–35 with healthy vision participated in the experimental measurement. These users embedded in the VR environment only had information about the active VR environment in which they could actively move. These users had no information about the color modification of the VR environment and immersed themselves in a VR art gallery with white walls. There was no noise or disturbing elements in the room intended for experimental measurements. In addition to the experimental measurement of physiological changes using polygraph sensors, the movements and reactions of users were also observed during immersion in the VR environment, and verbal reactions to the environment were recorded in writing. This study primarily focuses on the physiological changes measured by polygraph sensors; however, the fact that the participants noticed the first change in the color tone of the background of the VR environment at approximately the same time is already attractive for future research. However, it is necessary to repeat that attention was primarily focused on color modification intervals in absolute RGB value, and no other criteria and attributes were considered in the study. However, this fact predicts exciting possibilities in future research in the field of color and human vision in a VR environment, where it will also be possible to take into account the knowledge gained so far and thus include other methods of measuring the influence of the VR environment on users and data analysis, including evaluation methods based on machine learning. For these purposes, it will be necessary to collect a large amount of data, heralding larger groups of users divided by age since human vision and perception change throughout a person’s life. Here, it will be necessary to include vision defects such as nearsightedness or farsightedness. At the same time, it is possible to consider creating a control group without color variations for more distinct changes within the framework of measuring human reactions to color changes in a VR environment.

The Polygraph LX6 from the Lafayette Instrument POLYGRAPH company was used in the research. The SW Lafayette Polygraph System 11.8.5 was used to record sensor signals and their primary output. A user embedded in a VR environment of the virtual art gallery was scanned by the following sensor types: P1 Abdominal Respiration trace; P2 Thoracic Respiration trace; PL Photoelectric Plethysmograph; SE Activity Sensor; and GS Electrodermal Activity (EDA). These sensors are specified in more detail in Section 3. As part of the discussion, it is necessary to mention that basic signal measurement of these sensors was carried out. A separate analysis of signals from these sensors and their other attributes represents another challenge for future research concerning color and human perception in a VR environment.

The results of this research are presented in Section 3 and Section 4. The section dealing with the programming part of the color modification experiment in the VR environment describes the creation of a #C Script procedure created directly for this research. This pivotal part of the research is also the basis for the following measurement of physiological changes in the recorded physiological sensors. Due to the many possibilities of analysis, processing, and interpretation of the obtained data from the lie detector, extensive research was carried out on these possibilities. Currently, the subject of scientific research is the various processing options rather than just the usual statistical methods of data set processing [45,46]. In this work, however, classical statistical procedures were used to analyze lie detector data to find an essential procedure for this issue. In the time interval of 18 min, the total minimum value ^a^T_min_ and maximum value ^b^T_max_ for individual sensors were determined. The next step was the definition of intervals for individual modifications of absolute RGB colors: G(255) G_min_|G_max_; R(255) R_min_|R_max_; and B(255) B_min_|B_max_. All of these values are given in Table 2 in this text. These values are also shown graphically in Figure 10, which visually shows the differences in these values in the intervals of the individual displayed RGB colors and the total interval of the ^a^T_min_ and ^b^T_max_ of all lie detector sensors.

The differences in values are shown in Table 3. In Figure 10, the values of the GS Electrodermal Activity (EDA) sensor is not shown in Figure 7. The measured values of this sensor differ significantly from others. “Time series statistical methods” and recording evaluation were used for this sensor, which proved to be more suitable in the case of this sensor. Subsequently, a linear continuous trend was calculated: GS: y = 58,461x + 4 × 10^6^. This procedure of evaluating the measured data is illustrated in Section 4 in Figure 11.

The above procedures of data analysis and their interpretation proved to be suitable for evaluating the data in this experiment. This text describes the signal evaluation method for three intervals of sudden color change in a virtual gallery environment. This process makes it possible to evaluate such smooth color transitions in the VR environment’s total time interval of color changes: GS: y = 58,461x + 4 × 10^6^; P1: y = 2075.7x + 1 × 10^9^; P2: y = −44,509x + 2 × 10^9^; PL: y = 156.04x + 1 × 10^9^; and SE: y = −2021.5x + 7 × 10^8^. These linear trends for all signal recordings from individual sensors allow the user’s reactions to individual changes in color tones to be evaluated according to their points on the recorded curve. These curves are shown in Figure 11, Figure 12 and Figure 13 for individual lie detector sensors. It is likely that with these processes, it will be possible to more easily analyze other attributes associated with the measured signals of individual sensors and the physiological changes related to this. These additional procedures and analyses will be the subject of subsequent research, for which this experiment provides the basis.

In conclusion, it can be stated that measuring and analyzing user reactions to changes in the color environment in VR is possible. In this experiment, a lie detector was used as a detector of the response to a change in the environment. This issue requires further research into color and human vision in a VR environment. This issue also calls for further research on sensors and signal processing. However, practical use in the field of research in medicine, psychology, physiotherapy, art, criminology, and forensic science can be assumed. At the same time, these procedures can also be applied in commercial use areas.

## Figures and Tables

**Figure 1 sensors-24-04926-f001:**
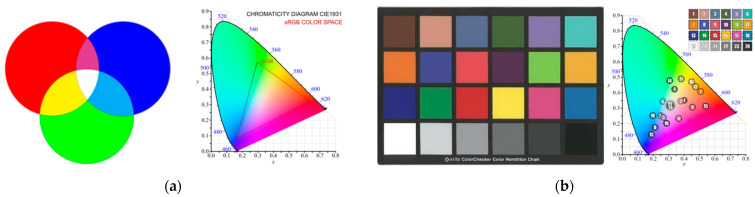
Color model and gamut: (**a**) RGB and color space sRGB in the Chromacity Diagram CIE 1931 and (**b**) standardized color scale and color position in the Chromacity Diagram CIE 1931 (CIE 1976) [21].

**Figure 2 sensors-24-04926-f002:**
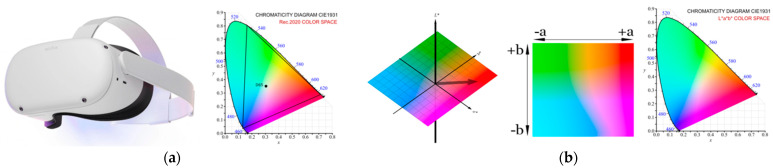
Color model and gamut: (**a**) VR headset Oculus Quest2 and color space Rec.2020 in the Chromacity Diagram CIE 1931 and (**b**) L*a*b* color model and L*a*b* gamut in the Chromacity Diagram CIE 1931.

**Figure 3 sensors-24-04926-f003:**
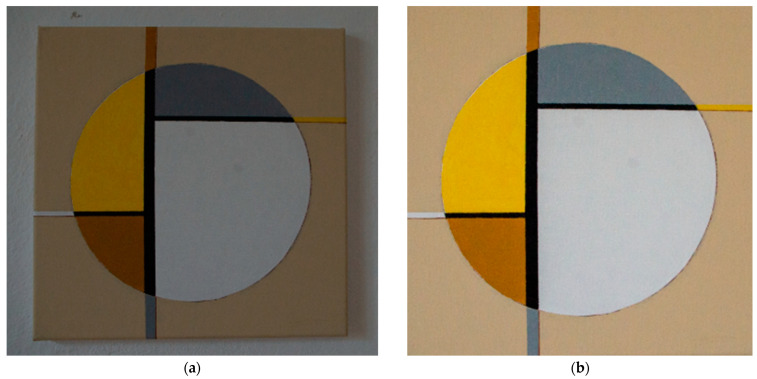
Digitization and creation of a digital twin of a work of art: (**a**) an original image in an art gallery environment and (**b**) a digitized art image for a VR environment.

**Figure 4 sensors-24-04926-f004:**
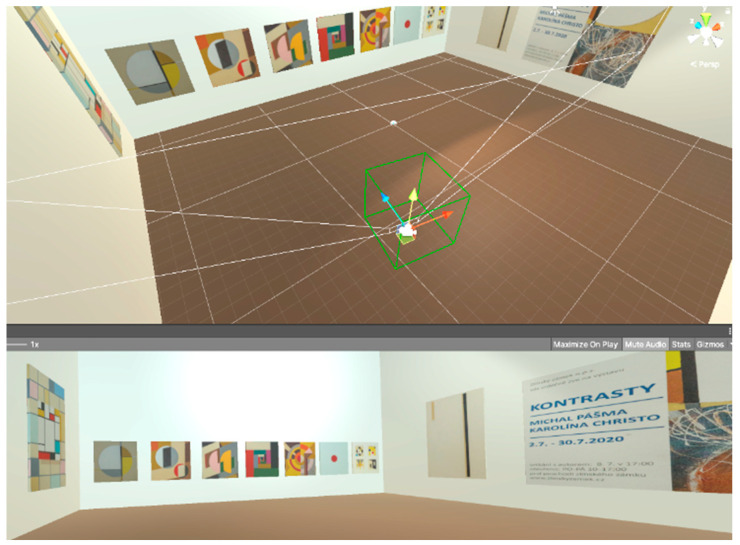
The virtual art gallery environment.

**Figure 5 sensors-24-04926-f005:**
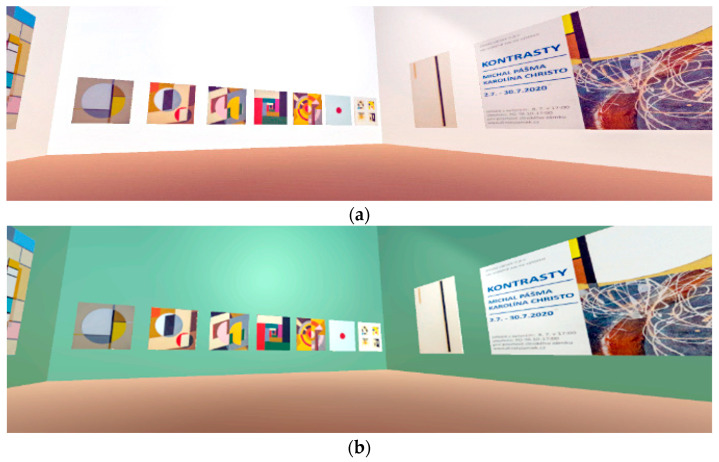
The initial white background of the VR environment: (**a**) white color is the first color to immerse the user in the VR environment before color modification and (**b**–**d**) reference images of the smooth color tone change in the VR gallery environment.

**Figure 6 sensors-24-04926-f006:**
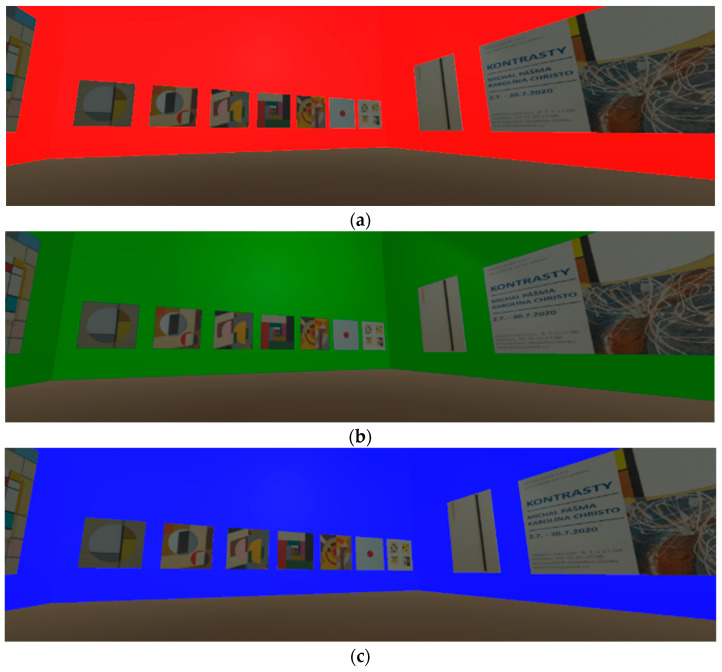
Static direct absolute RGB process colors: (**a**) direct absolute background color R (255), (**b**) direct absolute background color G (255), and (**c**) direct absolute background color B (255).

**Figure 7 sensors-24-04926-f007:**
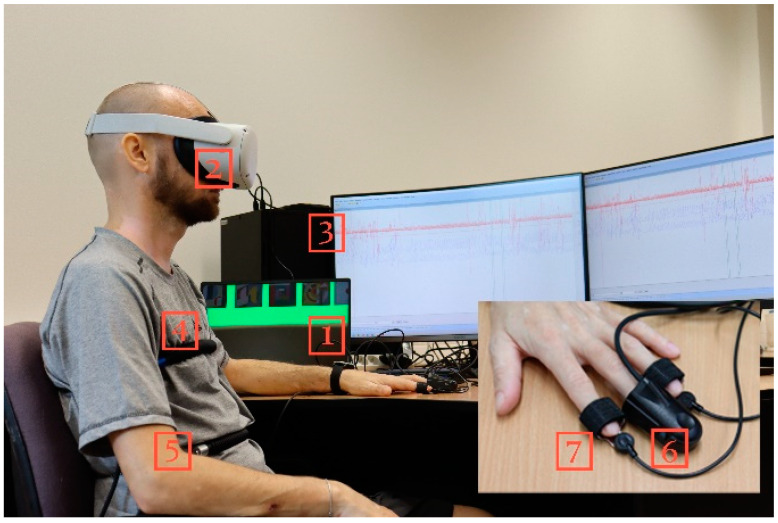
User connected to lie detector sensors in a virtual gallery environment: (1) and (3) computing and display units, (2) VR headset, (4) and (5) Pneumo Chest Assembly, (6) Photoelectric Plethysmograph, and (7) Electrodermal Activity (EDA).

**Figure 8 sensors-24-04926-f008:**
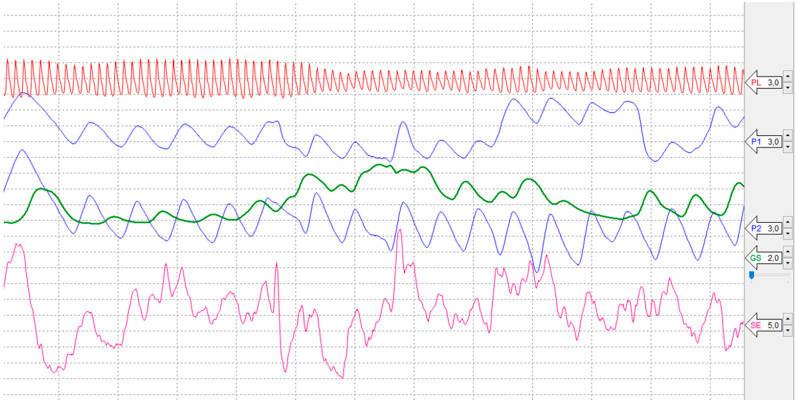
Detail of progress captured by lie detector sensors in a virtual gallery environment: (P1) and (P2) Pneumo Chest Assembly, (PL) Photoelectric Plethysmograph, (GS) Electrodermal Activity (EDA) and (SE) Activity Sensors.

**Figure 9 sensors-24-04926-f009:**
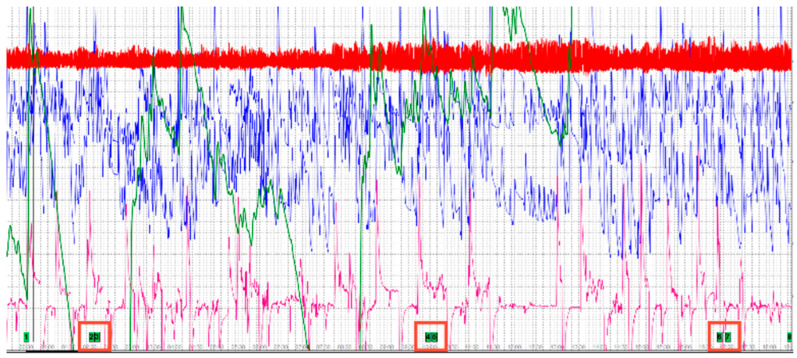
Display of lie detector signals measured by sensors in the total measurement time interval. Figure 8 shows the details of individual sensors.

**Figure 10 sensors-24-04926-f010:**
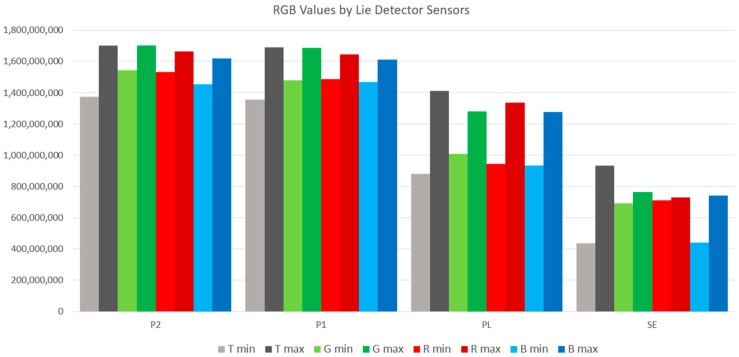
Graphic representation of signals measured by polygraph sensors.

**Figure 11 sensors-24-04926-f011:**
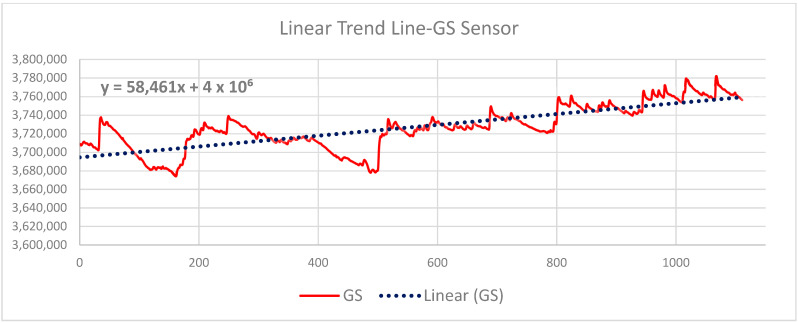
Graphic representation of the GS sensor signal (EDA).

**Figure 12 sensors-24-04926-f012:**
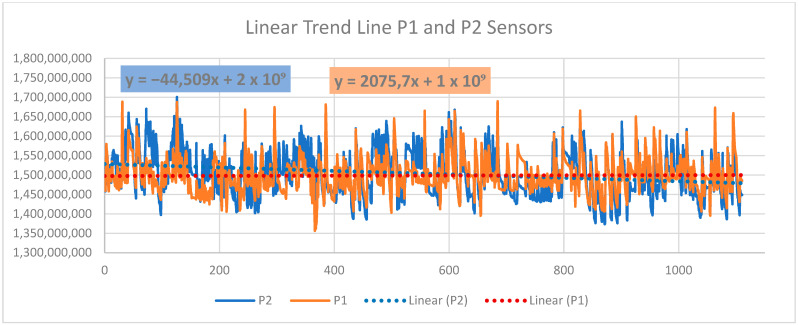
Graphic representation of the signal from the sensor P1 Abdominal Respiration trace and P2 Thoracic Respiration trace.

**Figure 13 sensors-24-04926-f013:**
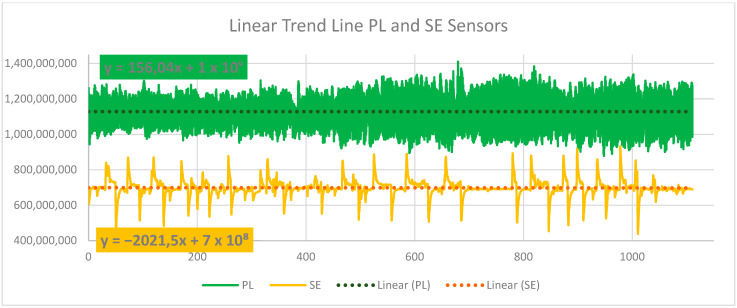
Graphic representation of the signal from the PL Photoelectric Plethysmograph and SE Activity Sensor sensors.

**Table 1 sensors-24-04926-t001:** RGB and L*a*b* color value.

Color	RGB Value	L*a*b* Value
Red	(255, 0, 0)	(53.2, 80.1, 67.2)
Green	(0, 255, 0)	(87.7, −86.2, 83.2)
Blue	(0, 0, 255)	(32.3, 79.2, −07.8)
White	(255, 255, 255)	(−00, 0.0, 0, −0.0)
Green Lima	(108, 203, 23)	(73.5, −54.4, 69.8)
Magenta Plum	(221, 160, 221)	(73.3, 32.5, −21.9)
Magenta	(255, 0, 255)	(60.3, 98.2, −60.8)
Mgenta Purple	(186, 143, 244)	(66.9, 35.9, −44.7)
Blue Purple	(132, 130, 230)	(58.5, 25.5, −50.5)

**Table 2 sensors-24-04926-t002:** Polygraph sensor signal values.

Color	Polygraph Sensors				
	P2	P1	PL	SE	GS
^a^T_min_	1,373,308,581	1,356,380,146	878,852,659	435,142,537	3,674,121
^b^T_max_	1,700,740,365	1,689,799,170	1,409,655,943	932,446,398	3,781,921
G _min_	1,544,195,568	1,480,290,055	1,007,992,714	693,660,008	3,681,374
G _max_	1,700,740,365	1,687,747,380	1,278,472,078	765,109,824	3,684,355
R _min_	1,530,578,442	1,484,640,044	942,374,819	711,869,260	3,729,730
R _max_	1,661,776,075	1,643,523,274	1,336,146,189	730,070,195	3,733,444
B _min_	1,453,597,565	1,467,732,400	934,089,157	438,385,740	3,753,788
B _max_	1,618,014,160	1,610,575,557	1,273,883,604	741,224,476	3,779,356

**Table 3 sensors-24-04926-t003:** Differences in the values of RGB and ^a^T_min_|^b^T_max_.

Color	Polygraph Sensors				
	P2	P1	PL	SE	GS
^a^T_min_	1,373,308,581	1,356,380,146	878,852,659	435,142,537	3,674,121
^b^T_max_	1,700,740,365	1,689,799,170	1,409,655,943	932,446,398	3,781,921
G _min_	1,544,195,568	1,480,290,055	1,007,992,714	693,660,008	3,681,374
G _max_	1,700,740,365	1,687,747,380	1,278,472,078	765,109,824	3,684,355
R _min_	1,530,578,442	1,484,640,044	942,374,819	711,869,260	3,729,730
R _max_	1,661,776,075	1,643,523,274	1,336,146,189	730,070,195	3,733,444
** B _min_ **	1,453,597,565	1 467,732,400	934,089,157	438,385,740	3,753,788
** B _max_ **	1,618,014,160	1,610,575,557	1,273,883,604	741,224,476	3,779,356

## Data Availability

The data presented in this study are available on request from the author (I.D.).

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
