# Peer review of "RGB Color Model: Effect of Color Change on a User in a VR Art Gallery Using Polygraph"

_sensors, 2024, doi:10.3390/s24154926_

Round 1

Reviewer 1 Report

Comments and Suggestions for Authors

This paper presents an experimental method for measuring the physiological response (respiration, blood pressure, etc.) in response to ambient colour change in a VR environment. Specifically, the authors set up a VR art gallery with digital replicas of art pieces. During the the experiment, the wall colour underwent gradual and sudden change, while the sensors recorded a range of physiological data.

The topic of this paper is interesting, as colour perception experiments are often behavioural, cognitive or low-level biological. The introduction of a polygraph and the use of a wider range of sensors makes an interesting proposal.

However, I have concerns about the impact and the technical contributions of the paper. Most of these could be addressed with improved presentation (e.g. explaining the exact hypothesis/research question that the experiment is trying to investigate), but the scope and range of contributions might not warrant a full paper.

Questions / concerns:

1. What is the exact contribution of this experiment? Is it just a proof of concept showing that such a study is possible (if so, it could be re-focussed to discuss methodological choices rather than a vague description of colour theory)? The paper should be better-structured and more focussed.

2. Why is a lie detector a feasible / preferrable method to detect responses compared to classical behavioural experiments (e.g. quality assessment / comparison / adjustment experiments in psychophysics)? The authors might want to discuss the advantages and disadvantages.

3. Have the authors attempted to calibrate the sensitivity of their proposed method (sensor-based response measurements) compared to classical behavioural experiments (e.g. physiological responses to change vs. conscious recognition of the change)?

4. The discussion of colour spaces should be reduced and substantially improved. The inclusion of CMYK seems unnecessary (unless it's used in the experimental protocol in a non-obvious way), and could be removed. However, the authors could try to include a discussion of some better-standardised colour spaces (CIE XYZ/Lab), as the RGB colour space on the Quest2 is unlikely to follow sRGB. Admitting this will help the reader to understand a limitation of the experimental protocol: in different VR headsets, the same RGB values correspond to different perceived colours (different XYZ/Lab values). Measuring the display profile of the VR headset first would be preferred.

5. Related to the previous note: colour interpolation would be better in Lab or a similar perceptually uniform colour space. Averaging/interpolating colours in RGB results in discolouration.

6. Experimental protocol: How many people took part in the experiment? How reliable were the results? Was there any attempt made to randomise the order of the cyclic colours or the time of the "three sudden intervals" to negate timing, fatigue and other effects? How did you pick the slowly-changing background colours (green/purple/pink)? With the 'very contrasting change', (160), did you also change the scene lighting? The art work seems substantially darker on these screenshots.

7. line 253: `Update()`, variable increased by 1: isn't this relying on a perfect frame rate? Wouldn't FixedUpdate work better in this case, or an increment proportionate to Time.deltaTime within update?

8. What are color models, and how are they different from color spaces?

9. line 306: how do you estimate the frequency of breathing (it's admittedly challenging)

Minor issues / presentation:

30-31: "At the same time, Computer graphics and Image processing use differ in each 30 field and individual subdisciplines." ambiguous sentence

"Therefore, it is necessary to recalculate these color models": therefore?

158: "indistinguishable from the human eye"

357: "RGB color model does not work with physical color but with colored light": what is "physical color" in this context? Are you referring to the difference between a light spectrum vs. Perceived colour?

461: HSB and HSV are somewhat closer aligned to perception, but there are clearly better alternatives, such as LMS/Lab

506: #C->C#

Typographic suggestions:

More common to cite before full stop [1]. Rather than after. [1]

101, 103: "from the 0.1 range" -> 0..1 or 0---1

Comments on the Quality of English Language

The paper was written with good use of English. Issues with the presentation are not linguistic.

Author Response

Dear Reviewer, thank you for your comments and attention.

  1. What is the exact contribution of this experiment? Is it just a proof of concept showing that such a study is possible (if so, it could be re-focussed to discuss methodological choices rather than a vague description of colour theory)? The paper should be better-structured and more focussed. The structure has been modified. Some methodologies were also added to the discussion as part of future research.

2. Why is a lie detector a feasible / preferrable method to detect responses compared to classical behavioural experiments (e.g. quality assessment / comparison / adjustment experiments in psychophysics)? The authors might want to discuss the advantages and disadvantages. Polygraph was chosen due to the absence of measurement of the influence of colors in terms of physiological changes. Future research envisages the involvement of other devices and methodologies.

3. Have the authors attempted to calibrate the sensitivity of their proposed method (sensor-based response measurements) compared to classical behavioural experiments (e.g. physiological responses to change vs. conscious recognition of the change)? Bylo provedeno pět testovacích měření za účelem kalibrace zařízení a testování senzorů. (330) Byli jsme během byli respondenti vizuálně pozorováno měření (pohyb, mimika). (330)

4. The discussion of colour spaces should be reduced and substantially improved. The inclusion of CMYK seems unnecessary (unless it's used in the experimental protocol in a non-obvious way), and could be removed. However, the authors could try to include a discussion of some better-standardised colour spaces (CIE XYZ/Lab), as the RGB colour space on the Quest2 is unlikely to follow sRGB. Admitting this will help the reader to understand a limitation of the experimental protocol: in different VR headsets, the same RGB values correspond to different perceived colours (different XYZ/Lab values). Measuring the display profile of the VR headset first would be preferred. Overall, section 2 (colorimetry) and the structure of this section were modified and supplemented.

5. Related to the previous note: colour interpolation would be better in Lab or a similar perceptually uniform colour space. Averaging/interpolating colours in RGB results in discolouration. Overall, section 2 (colorimetry) and the structure of this section were modified and supplemented.

6. Experimental protocol: How many people took part in the experiment? How reliable were the results? Was there any attempt made to randomise the order of the cyclic colours or the time of the "three sudden intervals" to negate timing, fatigue and other effects? How did you pick the slowly-changing background colours (green/purple/pink)? With the 'very contrasting change', (160), did you also change the scene lighting? The art work seems substantially darker on these screenshots. Section 2 and (206, 330) 

7. line 253: `Update()`, variable increased by 1: isn't this relying on a perfect frame rate? Wouldn't FixedUpdate work better in this case, or an increment proportionate to Time.deltaTime within update? Thank you for the useful suggestion for methodic. This is a change of 1 tonal value from the specified background colors.

8. What are color models, and how are they different from color spaces? Added to Section 2.

9. line 306: how do you estimate the frequency of breathing (it's admittedly challenging). These are demanding sensors. The measure was supplemented by visual observation and recording of verbal responses. (347 and 348)

Minor issues / presentation: All comments were corrected and the full text of the manuscript was formally reviewed.

30-31: "At the same time, Computer graphics and Image processing use differ in each 30 field and individual subdisciplines." ambiguous sentence •

"Therefore, it is necessary to recalculate these color models": therefore?

158: "indistinguishable from the human eye"

357: "RGB color model does not work with physical color but with colored light": what is "physical color" in this context? Are you referring to the difference between a light spectrum vs. Perceived colour?

461: HSB and HSV are somewhat closer aligned to perception, but there are clearly better alternatives, such as LMS/Lab Added to the discussion

506: #C->C#

Typographic suggestions:

More common to cite before full stop [1]. Rather than after. [1]

101, 103: "from the 0.1 range" -> 0..1 or 0---1

Reviewer 2 Report

Comments and Suggestions for Authors

The paper explores the influence of color variations on user perception within a virtual reality art gallery, employing an interdisciplinary approach that integrates methodologies from computer science, art, and psychology. However, there is a need to refine the methodology and experimental sections.

1. Ensure the provision of precise details regarding the participants, such as the total number, age bracket, and gender distribution, to facilitate a comprehensive understanding of the sample's representative nature.

2. Enrich the description of the experimental setup by outlining the laboratory configuration, lighting specifications, and noise mitigation measures to uphold the reliability of the experimental conditions.

3. Introducing a control group, devoid of color variations, is recommended to elucidate the distinct effects of color variations on physiological responses.

4. Provide elaborate explanations regarding participants' interactions within the VR environment, encompassing specific tasks, interactive instruments, and operational procedures.

Comments on the Quality of English Language

The author's English expression is clear and precise, effectively conveying meaning. To further enhance clarity, it is recommended that the author refine sentence structures by incorporating complex and compound sentences, thereby enriching the text's expressive impact.

Author Response

Dear Reviewer, thank you for your comments. 

  1. Ensure the provision of precise details regarding the participants, such as the total number, age bracket, and gender distribution, to facilitate a comprehensive understanding of the sample's representative nature. Added in the manuscript (330)
  2. Enrich the description of the experimental setup by outlining the laboratory configuration, lighting specifications, and noise mitigation measures to uphold the reliability of the experimental conditions. Added in the manuscript (330)
  3. Introducing a control group, devoid of color variations, is recommended to elucidate the distinct effects of color variations on physiological responses. Thank you for an excellent stimulus that will be very beneficial for future research and data analysis.
  4. Provide elaborate explanations regarding participants' interactions within the VR environment, encompassing specific tasks, interactive instruments, and operational procedures. Added in the manuscript Sections 2 and 3.

Round 2

Reviewer 2 Report

Comments and Suggestions for Authors

所有问题都已解决,准备出版。